# Testosterone Modulates Oxidative Stress in a Sexually Dimorphic Manner in CBA/Ca Mice Infected with *Plasmodium berghei* ANKA

**DOI:** 10.3390/ijms26083898

**Published:** 2025-04-20

**Authors:** Teresita de Jesús Nolasco-Pérez, Víctor Hugo Salazar-Castañón, Luis Antonio Cervantes-Candelas, Fidel Orlando Buendía-González, Jesús Aguilar-Castro, Martha Legorreta-Herrera

**Affiliations:** 1Laboratorio de Inmunología Molecular, Unidad de Investigación Química Computacional, Síntesis y Farmacología en Moléculas de Interés Biológico, División de Estudios de Posgrado e Investigación, Facultad de Estudios Superiores Zaragoza, Universidad Nacional Autónoma de México (UNAM), Ciudad de México 09320, CP, Mexico; teresyta.qfb@gmail.com (T.d.J.N.-P.); mestilom@hotmail.com (V.H.S.-C.); cervantescandelasluis@gmail.com (L.A.C.-C.); fidelbuendia2@gmail.com (F.O.B.-G.); jesus_aguilar_castro@yahoo.com.mx (J.A.-C.); 2Posgrado en Ciencias Biológicas, Unidad de Posgrado, Universidad Nacional Autónoma de México, Circuito de Posgrados, Ciudad Universitaria, Ciudad de México 04510, CP, Mexico

**Keywords:** *Plasmodium berghei* ANKA, malaria, testosterone, letrozole, oxidative stress, androgen, estrogen, superoxide dismutase, catalase, glutathione peroxidase

## Abstract

Malaria, the deadliest parasitic disease in the world, is sexually dimorphic, inflammatory, and oxidative. Males experience more severe symptoms and mortality than females do; therefore, the roles of 17β-estradiol and testosterone in this phenomenon have been studied. Both hormones affect oxidative stress, the primary mechanism of *Plasmodium* elimination. Estradiol has antioxidant activity, but the role of testosterone is controversial. Testosterone increases oxidative stress by reducing superoxide dismutase (SOD), glutathione peroxidase (GPx), and catalase (CAT) activities, which increase lipoperoxidation in the testis. However, the antioxidant properties of testosterone in prostate and nervous tissue have also been reported. The discrepancies are probably because when testosterone levels increase, the aromatase enzyme transforms testosterone into estrogens that possess antioxidant activity, which masks the results. Therefore, it is unknown whether testosterone is involved in the sexual dimorphism that occurs in oxidative stress in malaria. In this work, we administered testosterone and simultaneously inhibited aromatase with letrozole to evaluate the role of testosterone in the sexually dimorphic pattern of oxidative stress that occurs in the blood, spleen, and brain of male and female CBA/Ca mice infected with *Plasmodium berghei* ANKA (*P. berghei* ANKA). Testosterone triggers parasitemia in males, who also display more oxidative stress than females in the absence of infection, leading to sexually dimorphic patterns. Interestingly, increasing testosterone levels in infected mice reduced oxidative stress in males and increased oxidative stress in females, reversing or eliminating the dimorphic patterns observed. Oxidative stress varies in each tissue; the brain was the most protected, while the blood was the greatest damaged. Our findings highlight the role of testosterone as a regulator of oxidative stress in a tissue and sex-specific manner; therefore, understanding the role of testosterone in malaria may contribute to the development of sex-specific personalized antimalarial therapies.

## 1. Introduction

Malaria is the deadliest parasitic disease in the world [1]; it is sexually dimorphic, inflammatory, and oxidative [2,3], and males suffer more severe symptoms and mortality than females do at a ratio of 3:1, which peaks at the age of 21 to 30 years [4,5]. This sexual dimorphism has been attributed at least in part to the sex steroids 17β-estradiol and testosterone, as they are responsible for the most important physical and physiological differences between the sexes. Both hormones affect the immune response in malaria and oxidative stress [6,7], which is the primary mechanism of parasite elimination [8]. The sex hormone 17β-estradiol has the highest concentration in females, positively modulates the immune response, and has antioxidant activity [9]; in contrast, testosterone is the most concentrated androgen in males and possesses immunosuppressive properties [3]. However, its involvement in oxidative stress is controversial; testosterone has been described to increase oxidative stress, because it decreases the activities of the antioxidant enzymes superoxide dismutase (SOD), glutathione peroxidase (GPx), and catalase (CAT), which increase membrane lipoperoxidation in the testis [10,11,12]; conversely, it has also been reported that testosterone possesses antioxidant properties in prostate and nervous tissue [13,14]. Furthermore, increasing testosterone concentrations in vitro increases CAT activity in a twofold manner in an androgen receptor-dependent mechanism [13]. These results may differ, because the aromatase enzyme triggers an increase in estrogen levels as testosterone levels increase [15].

It has been shown that male and female mice infected with *P. berghei* ANKA differ in oxidative stress; after gonadectomy, female mice have reduced CAT, SOD, and GPx activities in the blood and spleen, which increases both malondialdehyde (MDA) levels and parasitemia, whereas the opposite occurs in male mice under the same conditions [7]. This is important in malaria, because macrophages, dendritic cells, and neutrophils possess specific receptors for testosterone [16,17]. These cells are specialists in eliminating *Plasmodium* via phagocytosis [18]. In that process, they generate reactive oxygen species (ROS), including superoxide anions (O_2_^−^), hydrogen peroxide (H_2_O_2_), hydroxyl radical (^·^OH), and nitric oxide radical (NO), and are considered to be ROS that react with O_2_^−·^ to produce peroxynitrite (ONOO^−^), which is highly reactive and from which nitrate (NO_3_^−^) and nitrite (NO_2_^−^) are derived. These molecules are known as reactive nitrogen species (RNS) [19]. Nitric oxide causes *Plasmodium* death in vitro [20] and is related to the immunomodulation of the immune response in malaria in vivo [21]. Both ROS and RNS oxidize the main macromolecules, generating oxidative stress that damages the parasite. When these molecules are in excess, they also induce inflammation and damage to the host, resulting in pathology [22,23]. Another source of free radicals in malaria is hemoglobin [24], which the parasite metabolizes to obtain amino acids and nutrients. These then release the heme group with Fe^3+^, which, in the presence of O_2_, generates hem Fe^2+^ and O_2_^−·^, the substrate of SOD, which transforms O_2_^−·^ into hydrogen peroxide, H_2_O_2_. In addition, O_2_^−·^ together with the hem Fe^2+^ previously generated, produces the hydroxyl radical (^·^OH) via the Fenton reaction, which creates oxidative stress [25]. In response to the oxidative stress caused by phagocytes and *Plasmodium* metabolism, cells possess a complex network of antioxidant compounds, such as glutathione, and enzymes, such as superoxide dismutase (SOD), catalase (CAT), and glutathione peroxidase (GPx) [26,27,28]. However, whether testosterone is involved in the dimorphic oxidative stress that occurs in malaria has not been studied. We hypothesize that increasing the concentration of testosterone and simultaneously inhibiting aromatase to avoid estrogen interference will allow us to determine the contribution of testosterone in the dimorphic pattern that occurs in malaria-induced oxidative stress. We expect testosterone to increase parasitemia in both sexes and increase oxidative stress in males. In this work, we analyzed the impact of experimentally increasing testosterone concentrations and simultaneously inhibiting aromatase in vivo via letrozole (a highly selective inhibitor of aromatase, to prevent estrogen interference [29]) in female and male mice to study the sexually dimorphic pattern of oxidative stress in mice infected with *P. berghei* ANKA. We studied the most affected tissues in this disease, such as the blood, spleen, and brain. In the blood, *Plasmodium* induces oxidative stress in both infected and noninfected erythrocytes, which promotes their elimination and causes anemia [30]; the spleen plays a central role in malaria parasite removal [31]; and the brain possesses a high lipid content that makes it more susceptible to oxidative stress, which is involved in cerebral malaria, the most lethal complication of the disease [32]. In addition, the levels of testosterone, estradiol, and DHEA were quantified. To evaluate oxidative stress, the specific activities of SOD, GPx, and CAT and the concentrations of MDA, a marker of lipoperoxidation, and nitric oxide (NO), measured as nitrites (NO_2_^−^) and nitrates (NO_3_^−^) via the Griess reaction, were also evaluated in the blood, spleen, and brain of male and female CBA/Ca mice.

To our knowledge, this is the first study to investigate the role of testosterone in the dimorphic pattern of oxidative stress in malaria in detail. This study is important, because it demonstrates that oxidative stress differs in different tissues related to malaria pathology. In addition, oxidative stress is the major mechanism that uses the immune response to eliminate the parasite, and the most important antimalarial drugs involve the generation of oxidative stress. Therefore, understanding the role of testosterone in oxidative stress in malaria could contribute to the development of personalized therapeutic approaches that consider the patient’s sex.

## 2. Results

### 2.1. The Combination of Letrozole and Testosterone Increased Testosterone Levels and Decreased 17β-Estradiol Levels in P. berghei ANKA-Infected Mice

To study the role of testosterone in the sexual dimorphism of malaria oxidative stress, we treated CBA/Ca mice with testosterone; to prevent testosterone from being converted to 17β-estradiol and estrone by the aromatase enzyme, we treated those mice with letrozole (a specific aromatase inhibitor) to avoid interference from both estrogens [33]. We first verified that the mixture of testosterone with letrozole increased testosterone levels in uninfected and *P. berghei* ANKA-infected mice. Interestingly, infection decreased free testosterone levels in both sexes; in addition, infected males treated with letrozole or testosterone presented increased free testosterone concentrations, which generated sexually dimorphic patterns. Finally, the combination of letrozole and testosterone significantly increased free testosterone levels in both sexes, eliminating the sexually dimorphic pattern (Figure 1A).

Relative to the 17β-estradiol levels, the combination of letrozole with testosterone decreased the 17β-estradiol concentration in uninfected male mice (Figure 1B). In addition, the levels of 17β-estradiol decreased in infected females treated with letrozole, and the levels of 17β-estradiol decreased in male mice treated with the combination of letrozole and testosterone (Figure 1B).

Additionally, we quantified the concentration of DHEA, a precursor of both testosterone and 17β-estradiol. DHEA levels in all groups of uninfected females were significantly lower than those in males under the same conditions, which generated dimorphic patterns. Interestingly, infection increased the levels of DHEA only in females, eliminating the sexual pattern in all groups (Figure 1C).

### 2.2. Increasing Testosterone Concentrations Increased Parasitemia in Both Sexes but Affected Males to a Greater Extent

Once we demonstrated that testosterone and the combination of letrozole with testosterone increased the free testosterone concentration (biologically active), we assessed parasitemia. Letrozole increased parasitemia in females (Figure 2A). As the testosterone concentration increased, parasitemia increased in both sexes (Figure 2A,B). Nevertheless, except for the group of females that received letrozole on day 6, the parasitemia of males was greater than that of females in all groups, which generated dimorphic patterns, particularly on days 7 and 8 post infection (Figure 2C).

### 2.3. Effects of Increasing Testosterone Concentration on Oxidative Stress in the Blood of P. berghei ANKA-Infected Mice

Males infected with *Plasmodium* exhibit more severe symptoms and mortality than females do [34]. Testosterone is the most important sex hormone in males, and the primary mechanism of the elimination of *Plasmodium* parasites is oxidative stress [35]. We studied whether testosterone is involved in the sexual dimorphism detected in oxidative stress in experimental cerebral malaria [7].

The first tissue we analyzed was blood, and we measured the specific activities of the enzymes SOD, GPx, and CAT. In addition, we examined the levels of the oxidative stress markers MDA, nitrite (NO_2_^−^), and nitrate (NO_3_^−^) as indirect measures of nitric oxide (NO) levels in infected female and male mice treated with both testosterone and letrozole and infected with *P. berghei* ANKA.

In this work, all groups of uninfected male mice presented greater SOD activity than females did, indicating a markedly sexually dimorphic pattern; furthermore, compared with the control untreated mice, the group of males receiving the combination of letrozole and testosterone presented decreased activity of this enzyme. Interestingly, infection increased enzyme activity only in the female groups, which inverted the previously described dimorphic pattern in all groups of uninfected mice, and a decreasing trend in SOD activity was detected as the concentration of free testosterone increased in both sexes (Figure 3A).

With respect to GPx and CAT activity in uninfected animals, a clear dimorphic pattern was detected in which uninfected males presented higher activity of both enzymes than females did. Infection decreased the activity of GPx and CAT in males, but in females, GPx activity was not modified, and CAT activity decreased depending on the concentration of testosterone, eliminating all dimorphic patterns detected in uninfected mice except for CAT activity in the untreated infected group, in which females presented greater activity than males did and exhibited a sexually dimorphic pattern. Interestingly, as the testosterone levels increased, the specific activity of both enzymes decreased (Figure 3B,C). In addition, to evaluate oxidative stress damage to lipids and tissues, we quantified the MDA, nitrite (NO_2_^−^), and nitrate (NO_3_^−^) contents as indirect measures of NO levels in each tissue. All groups of uninfected male mice presented higher MDA concentrations than females did under all conditions, which generated dimorphic patterns in all the uninfected groups. However, infection increased the MDA concentration in females but decreased it in males, eliminating the previously described sexually dimorphic patterns (Figure 3D). As expected, the infected testosterone-treated females presented greater lipoperoxidation than the infected-untreated control females did MDA levels, as expected. However, treatment with testosterone or infection did not significantly modify the blood NO_2_^−^ and NO_3_^−^ (NO) concentrations (Figure 3E).

### 2.4. Effects of Increasing Testosterone Concentrations on Oxidative Stress in the Spleens of P. berghei ANKA-Infected Mice

Since the spleen is the main site of parasite elimination in malaria [31], we analyzed the antioxidant activity and oxidative stress in this organ. SOD activity in the uninfected groups was extremely low; however, males presented greater activity than females did, but the difference was significant only in the untreated control group and in the group treated with the combination of testosterone and letrozole, which exhibited a sexually dimorphic pattern. Infection increased SOD activity only in males treated with the combination of testosterone and letrozole (Figure 4A). Relative to GPx, testosterone increased this enzyme activity in uninfected male mice, generating a sexually dimorphic pattern. Infection decreased GPx activity in all the groups and eliminated the dimorphic pattern described above (Figure 4B). With respect to CAT activity, uninfected males exhibited greater activity than females under the same conditions; increasing the testosterone concentration in uninfected mice increased CAT activity only in males, generating a sexually dimorphic pattern. In addition, infection reduced catalase activity in all groups, eliminating the dimorphic patterns described in uninfected mice (Figure 4C). The MDA concentration in uninfected mice was greater in males than in females, indicating sexually dimorphic patterns under all conditions. Infection increased the MDA concentration exclusively in females, eliminating the dimorphic patterns described above (Figure 4D).

Finally, NO_2_^−^ and NO_3_^−^ as indirect measures of the NO concentration in uninfected mice were greater in males than in females, but the difference was not significant. However, in infected mice, the NO concentration decreased in males, whereas in females, it did not change (Figure 4E).

### 2.5. Effects of Increasing Testosterone Concentration on Oxidative Stress in the Brains of P. berghei ANKA-Infected Mice

SOD activity in uninfected mice was greater in the brains of males than in those of females, indicating a dimorphic pattern under all conditions. Infection increased the activity of this enzyme exclusively in males; this increase was a function of testosterone concentration, and a dimorphic pattern was detected in the infected groups treated with the combination of letrozole and testosterone (Figure 5A). Infection decreased GPx activity in males, eliminating the dimorphic patterns described in uninfected mice (Figure 5B).

With respect to CAT activity, uninfected males presented higher levels than females did, but the difference was significant only in the groups treated with letrozole or testosterone alone, which generated sexually dimorphic patterns. Infection decreased the activity of CAT only in males, eliminating the dimorphic pattern described above (Figure 5C).

The MDA concentration was greater in uninfected males than in females under the same conditions, resulting in sexually dimorphic patterns in all groups. Infection eliminated the dimorphic patterns by decreasing the MDA levels in males (Figure 5D). With respect to NO_2_^−^ and nitrates (NO_3_^−^), an indirect measure of nitric oxide concentration (NO) in uninfected male mice presented higher NO levels than those in females did, indicating a dimorphic pattern except in the groups treated with the combination of letrozole and testosterone. Infection increased NO levels in females, which inverted the sexually dimorphic pattern in the group treated with testosterone, as it increased the NO concentration exclusively in females (Figure 5E).

## 3. Discussion

To study the role of testosterone in the immune response, its concentration is often increased [36]; however, it is not considered that this promotes its biotransformation into estrogens by the aromatase enzyme; this is important, because estrogens possess antioxidant properties due to their structure, which is independent of its interaction with the receptor [37]. In addition, estrogen provokes antioxidant effects that induce SOD and GPx antioxidant activity [38]. In this work, we inhibited aromatase via letrozole in *P. berghei* ANKA-infected CBA/Ca mice treated with testosterone to study the role of testosterone in the sexually dimorphic pattern of oxidative stress in *P. berghei* ANKA infection.

First, we verified that the combination of testosterone with letrozole increased testosterone levels and decreased 17β-estradiol concentrations. In addition, we detected that parasitemia increased in both sexes as the testosterone concentration increased. Furthermore, antioxidant enzyme activity was modified as a function of testosterone concentration; thus, oxidative stress differed by sex, infection, and in a tissue-dependent manner.

Interestingly, in all the uninfected groups, the concentration of the hormone DHEA (a precursor of testosterone and 17β-estradiol) was greater in males than in females, indicating a sexually dimorphic pattern; a probable explanation is that the adrenal glands of males produce more DHEA than those of females do, as described by Hobe’s group [39]. Interestingly, infection equaled DHEA concentrations in males and females regardless of testosterone or 17β-estradiol concentration; this finding is likely the result of the metabolic and stress changes induced by *Plasmodium* infection, particularly in females, since cortisol inhibits the enzyme 3β-hydroxysteroid dehydrogenase type 2, which blocks the conversion of DHEA to androstenedione, increasing DHEA levels as the cortisol concentration increases [40]. In addition, as the testosterone concentration increased, males developed greater parasitemia than females did, which generated dimorphic patterns. This finding confirms what was described by our group [6,41], and a likely explanation is that, as explained by Benten et al. [42], increasing levels of testosterone decrease both the concentration of IgG antibodies and the number of spleen cells by 50% in that organ, immune response cells eliminate the parasite [32], and the most recognized mechanism of parasite elimination is oxidative stress [8].

In this work, we analyzed whether testosterone modifies oxidative stress in the blood, spleen, and brain in a cerebral malaria model. In the blood, cells are continuously exposed to the parasite; the spleen is the primary site of *Plasmodium* destruction through the immune response via phagocytosis and oxidative stress [43]. Finally, we analyzed the brain, because oxidative damage is associated with cerebral malaria [44]. Blood was the tissue that experienced the greatest degree of oxidative stress. The greater activity of SOD, GPx, and CAT and increased levels of MDA in the blood of males than in those of females demonstrate that, in the absence of infection, males develop greater oxidative stress; a possible explanation is that testicular membranes are rich in polyunsaturated fatty acids and, therefore, are more susceptible to oxidative stress [45]. Another possible explanation is the metabolism of female mice, which produce less SOD and GPx than males do under normal conditions [46]. Interestingly, infection dramatically altered this pattern; in males, SOD, GPx, and CAT activities decreased, likely because the enzymes were consumed in an attempt to counteract the reactive species generated by the infection, corroborating the findings that patients with severe malaria decrease SOD activity [47], and as a consequence, GPx activity also decreases compared to that of uninfected individuals [48]. This phenomenon was also detected in BALB/c mice infected with *P. berghei* ANKA [11]. In this work, females presented increased SOD activity, which reversed the dimorphic pattern; this increase was probably sufficient to neutralize the superoxide anion generated by phagocytic cells, and therefore, neither GPx nor CAT activity was modified in that tissue.

Interestingly, the infected female group treated with testosterone presented reduced CAT activity, which explains the increase in both MDA and nitric oxide in that group. These findings demonstrate that females have greater antioxidant activity in their blood than males do, similar to what has been described in humans with heart disease [49,50,51]. Notably, MDA concentrations were lower in infected males than in uninfected controls; a likely explanation for this finding is that *Plasmodium* infection resulted in a deficiency in the respiratory burst of immune response cells, which reduced the number of oxidative species, as previously described [52,53]. However, this effect was not present in females, in whom increasing the testosterone concentration increased lipid peroxidation. This finding could be explained by the decrease in SOD and CAT activity in this group, corroborating that testosterone suppresses SOD activity under stress conditions [10].

When we analyzed the spleen, we found that uninfected females presented less oxidative stress than males did, as measured by lower SOD, GPx, and CAT activities and a lower MDA concentration, which generated a clear dimorphic pattern. These findings corroborate that, in general, uninfected females exhibit lower oxidative stress and ROS production than males do [49,54]. Estrogen, which has antioxidant properties, likely contributes to explain this finding. Furthermore, we found that administering testosterone increased GPx and CAT activity only in uninfected males. Contrary to our expectations, the combination of letrozole and testosterone did not affect the activity of either enzyme in the uninfected groups. It is likely that the increase, albeit minimal, in the 17β-estradiol levels in this group prevented changes in the MDA concentration. In addition, in infected mice, the combination of testosterone and letrozole increased SOD activity exclusively in males, possibly because their macrophages have a greater number of androgen receptors than females do [55]. Testosterone has also been shown to increase SOD activity [56]. However, the increase did not affect the concentration of MDA or NO in males. A likely explanation is that, since this group had the highest level of parasitemia, the superoxide radical level was increased, as previously described [57]. In addition, this group presented a high concentration of testosterone, a steroid that modulates the activity of macrophages in the spleen [58] and increases the activity of NADPH oxidase [59,60]. This enzyme synthesizes the superoxide radical (a substrate for SOD). The increase in SOD in this group was likely sufficient to neutralize reactive species, such as superoxide anions, generated by immune cells in response to infection; therefore, the activities of the GPx and CAT enzymes were not modified. In addition, infection decreased CAT activity in testosterone-treated males, probably because it could indirectly influence CAT expression by regulating T-cell and macrophage signaling pathways in the spleen [61]. Furthermore, testosterone is likely to increase mitochondrial activity [62,63] and, therefore, increase ROS production, thereby decreasing CAT activity. This occurs only in the spleen of males, probably because they have a greater number of androgen receptors than females do in the same tissue [64].

A clear dimorphic pattern was also detected in the brains of uninfected animals. Uninfected male mice presented greater activity of the enzymes SOD, GPx, and CAT, in addition to higher concentrations of MDA and NO than females did, which confirms that males in the absence of infection suffer from greater oxidative stress in the brain than females do. This is likely a consequence of the lower concentration of 17β-estradiol and the lower number of receptors for that hormone in males [65], because 17β-estradiol has antioxidant activity in the brain [66] and induces antioxidant signaling pathways [67]. On the other hand, our results are in agreement with those described by Wang and colleagues, who reported that testosterone increases the antioxidant activity of SOD in the brain, which may protect against brain damage caused by oxidative stress [68]. Notably, infection increased SOD activity exclusively in males in a manner dependent on the testosterone concentration in a similar way that it did in the spleen, which corroborates the findings described by Meydan and colleagues in the brain [69]. It is also likely that the high parasitemia induced the increase in SOD activity in this group in the manner described by Francischetti and colleagues and that SOD activity probably decreased the damage caused in the brain [70], as shown by the reduction in the MDA concentration. Another probable explanation is that females with lower parasitemia presented lower oxidative stress due to parasite metabolism, as previously reviewed [71,72]. Another possibility is that testosterone promotes the activation of the antioxidant response transcription factor Nrf2 in the brain [73]. In addition, testosterone can reduce the death of neurons in the hippocampus due to oxidative stress [74]. Furthermore, infection decreased CAT activity in the brain exclusively in males, which eliminated the dimorphic pattern observed in uninfected mice. In the absence of infection, CAT helps maintain neuroglial cells [75]. However, during *Plasmodium* infection, endothelial cells and macrophages increase the synthesis of H_2_O_2_ [76], the main substrate of CAT, which possibly decreased the enzyme activity. In addition, the activity of both cell types increases in the presence of testosterone [77,78], which partly explains what was observed in males. A summary of the main differences between males and females is shown in Table 1.

Our findings corroborate the antioxidant effects of testosterone in the brain under pathological conditions [14,79]. Therefore, evaluating whether females exhibit greater protection at the blood–brain barrier and, consequently, suffer less invasion of the brain by the parasite than males is crucial.

On the other hand, testosterone at supraphysiological doses decreases eNOS enzyme activity and NO formation, which reduces inflammation [80]. In this work, we found that administering testosterone significantly decreased the concentration of nitrites and nitrates, an indirect measure of NO in the brain, exclusively in males (Figure 6). This finding corresponds with that described by Marin et al., who demonstrated in vitro that the increase in nitric oxide levels is a function of testosterone concentration [81].

## 4. Materials and Methods

### 4.1. Mice

We used CBA/Ca mice infected with *P. berghei* ANKA, because this model is considered the gold standard for the study of cerebral malaria, which is the main complication that causes death [82], and because oxidative stress is directly involved in the pathogenesis of cerebral malaria [83]. The breeding units of CBA/Ca mice were initially donated by Dr. William Jarra (National Institute for Medical Research, London, UK). The mice were bred and maintained in a specific pathogen-free environment in the vivarium of the FES Zaragoza, UNAM. All the mice used in this work were female or male (12 weeks old) and were randomly grouped by sex. The animals were maintained within a double isolation barrier with a 12 h light/dark cycle, temperature ranging from 19.5 to 20 °C, filtered air, a sterile bed, sterile drinking water, and food ad libitum. Because the mouse breeding units were located in the same vivarium, we transferred the mice only from the breeding site to the experimental cubicle two weeks before the experiment, maintaining the same conditions described above.

The inclusion criteria were as follows: the mice were of the same strain, age, and sex in each cage.

### 4.2. Parasites and Infection

*P. berghei* ANKA was also a kind donation from Dr. W. Jarra; the parasite was cryopreserved in liquid nitrogen, and to activate it, a vial was thawed and ip inoculated into a 4-week-old mouse. When parasitemia reached 20%, blood was obtained and diluted in PBS, and an inoculum of 1 × 10^3^ parasitized erythrocytes/mL was prepared; 100 µL was injected iv into each mouse.

### 4.3. Parasitemia

Parasitemia was assessed daily via Giemsa-stained blood smears. The course of infection is shown as the geometric mean of the percentage of parasitemia in each group.

### 4.4. Testosterone and Letrozole Administration

The mice were treated with 30 mg/kg testosterone (Schering Plough, Newton, NY, USA) or vehicle (almond oil Zeta Farmaceutici, Brescia, Italy) subcutaneously every 72 h for 3 weeks, as described by Benten et al. [84]. The day after the last dose of testosterone, the mice were infected with 1 × 10^3^ parasitized erythrocytes.

Letrozole (Sigma-Aldrich, St. Louis, MO, USA) was administered, as previously described [41]. Briefly, letrozole was macerated in a mortar to decrease the particle size, after which a suspension was prepared with almond oil (vehicle). Each mouse was administered 7 mg/kg/30 µL daily for 14 days before infection and 6 days post infection [41].

### 4.5. Quantification of Testosterone, 17β-Estradiol, and DHEA

Sex steroids were quantified, as previously described [41]. Briefly, on day 8 post infection, all the mice were euthanized. Heart blood was recovered in heparinized tubes and centrifuged at 1000× *g* for 5 min. The plasma was separated and frozen at −20 °C until use. One hundred microliters of plasma was mixed with 5 mL of ethyl ether (JT Baker, Fisher Scientific SL, San Diego, CA, USA). The aqueous phase was frozen in a dry ice bath and mixed with ethanol (Sigma-Aldrich). The organic phase was transferred to a glass tube, and the ether was evaporated in a water bath for 48 h. The extract was rehydrated in a dry ice bath with ethanol (Sigma-Aldrich). The extract was rehydrated with 1000 µL of PBS/0.1% gelatin (Sigma-Aldrich).

The levels of free testosterone were quantified via the commercial method EIA-2924 (DRG International, Frauenbergstr, Marburg, Germany), following the instructions of the company. Briefly, 20 µL of the steroidal extract or standard was incubated with 100 µL of conjugate for 1 h at 37 °C. The plate was washed, 100 µL of substrate was added, and the mixture was incubated at 28 °C for 15 min. Stopping solution was added, and the absorbance was read at 620 nm on a Multiskan Ascent 96 plate reader (Thermo Fisher Scientific, Waltham, MA, USA).

The levels of 17β-estradiol were analyzed via a commercial Siemens Immulite LKE chemiluminescent immunoassay (Immulite 1000, Siemens Llanberis Gwynedd, Caernarfon, UK). Five hundred microliters of the steroidal extract was used and placed in the automated kit (Siemens Healthinneers, Surrey, UK).

The concentration of dehydroepiandrosterone (DHEA) was quantified via the commercial method DRG EIA-3415 (DRG). Ten microliters of each standard or steroid extract were mixed with 100 µL of the conjugate and incubated for 1 h at room temperature. The plate was washed and incubated with 100 µL of the substrate at 28 °C for 15 min. The reaction was halted with 100 µL of stop solution, and the absorbance was read at 630 nm with a Multiskan Go 96 plate reader (Thermo Fisher Scientific).

### 4.6. Quantification of SOD, GPx, and CAT-Specific Activities

The specific activity of SOD was quantified via the commercial method RANSOD (Randox Laboratories, Antrim, UK), as described previously [85]. Briefly, blood was collected in heparinized tubes. Erythrocytes or spleen or brain tissues were washed 4 times with isotonic NaCl solution and centrifuged. The cells were mixed with 1.0 mL of sterile distilled water and incubated at 4 °C for 15 min. The lysate was diluted 25-fold with 0.01 mmol/L phosphate buffer (pH 7.0), and 25 µL of this solution was mixed with xanthine oxidase substrate. The kinetics of the reaction were quantified at 505 nm and measured via a Thermo Scientific Multiskan Go plate spectrophotometer (Thermo Fischer Scientific, Abingdon, UK). Finally, the specific activity was reported as units/mg protein.

To identify the specific activity of glutathione peroxidase (GPx), the Ransel commercial method (Randox Laboratories) was used. Fifty microliters of heparinized blood or a suspension of spleen and brain tissues was used and diluted, according to the manufacturer’s instructions. Specific GPx activity was calculated as the decrease in absorbance, which was measured at 340 nm measured on a spectrophotometer (Thermo Scientific Multiskan Go) and reported as units/mg protein.

To the CAT-specific activity of erythrocytes or tissue suspensions was quantified via the method described by Aebi [86]. Briefly, the heparinized blood was centrifuged, the supernatant was removed, four parts sterile distilled water were added to the button and mixed, and the lysate was diluted 1:500 in phosphate buffer (pH 4). One microliter of the resulting solution was mixed with 500 µL of 30 mM hydrogen peroxide, and the absorbance was measured immediately at 240 nm (A1) and again at one minute (A2) via a UV plate spectrophotometer (Thermo Scientific Multiskan Go). The difference in absorbance per minute is a measure of enzyme activity. The results are presented as nmol H_2_O_2_ consumed per minute/mg protein.

### 4.7. Quantification of the MDA Concentration

It is common to measure the concentration of MDA to quantify the effects of free radicals on cells. This technique measures the lipoperoxidation of cell membranes and is a marker of oxidative stress [87]. In this work, the concentrations of MDA in the plasma, spleen, and brain were determined via a previously described method [85]. Briefly, homogenized spleen, brain, or plasma tissues corresponding to 1 mg of protein or standard (1,1,3,3-tetra methoxy propane (Sigma-Aldrich)) were mixed with 100 µL of orthophosphoric acid (0.2 M, Sigma-Aldrich), 125 µL of BHT (butylated hydroxytoluene 2 mmol/L; Sigma), and 12.5 µL of 0.1 M thiobarbituric acid (Fluka Chem, Buch, Switzerland; Sigma) dissolved in 0.11 M NaOH. The samples and standards were incubated at 90 °C for 45 min, immediately placed on ice and extracted with 250 µL of n-butanol (Sigma). The butanolic phase was separated by centrifugation at 1500× *g* for 3 min. The absorbance was measured at 535 nm on a plate reader (Thermo Fisher Multiskan Go). Finally, the concentration of MDA was calculated via a standard curve.

### 4.8. Nitrite and Nitrate Quantification as an Indirect Measure of NO Levels

The Griess method was used, as previously described [21]. Briefly, 15 µL of homogenized spleen, brain, or plasma tissue containing 1 mg of protein were incubated with 0.025 U of nitrate reductase (Sigma) and 75 µg of NADPH (Sigma) for 2 h at room temperature. One hundred microliters of Griess reagent (1% sulfanilamide, 0.1% N-(1-naphthyl)ethylenediamine dihydrochloride in orthophosphoric acid, Sigma) was added. Finally, 100 µL of trichloroacetic acid (10% in water) was added. The protein was removed by centrifugation, 100 µL of the supernatant was transferred to a 96-well plate, and the absorbance was measured at 540 nm in a plate reader (Thermo Scientific Multiskan Go).

### 4.9. Experimental Design

Five groups of 10 male and five groups of 10 female 12-week-old CBA/Ca mice were used. The first group of male mice was not treated (untreated), the second group was administered vehicle (vehicle), the third group was administered testosterone for 3 weeks, the fourth group was treated with letrozole, and the fifth group was administered a combination of letrozole and testosterone. One day before infection, every group was subdivided into two subgroups (n = 5): one subgroup was infected with *P. berghei* ANKA, and the other was injected with PBS and used as a negative control for infection; the same process was followed for female mice. All the mice were randomly grouped by sex and were euthanized on day 8 post infection. Blood, the spleen, and the brain were extracted from each mouse to evaluate oxidative stress and antioxidant activity, as described previously. The researchers were aware of the group allocation at all stages of the experiment and data analysis.

### 4.10. Statistical Analysis

The statistical design was planned according to data independence. Parasitemia was measured at specific times during infection in the same individual and was analyzed via two-way ANOVA with the Bonferroni post hoc test; the *p* value in each comparison is indicated in each graph (n = 5). The n = 5 was selected, because a larger number of individuals is experimentally unmanageable, and because 5 individuals ensure a reliable sample with 95% confidence.

The steroid concentration, specific enzyme activity of SOD, GPx, and CAT, and MDA and NO levels were analyzed via one-way ANOVA, because the data were obtained from groups independent of each other and are not time-dependent. The *p* value in each comparison is indicated in each graph (n = 5).

## 5. Conclusions

Testosterone induces a dimorphic pattern of oxidative stress that generally protects males from oxidative damage during *P. berghei* ANKA infection; however, the price is greater for parasitemia in males than in females. Oxidative stress is different in the blood, spleen, and brain; it also differs between males and females. Testosterone affects females only when the testosterone concentration is increased without aromatase inhibition, probably because this increases estrogen levels. Our findings are relevant, because they show for the first time that testosterone protects the brain against oxidative stress in male mice infected with *P. berghei* ANKA. Finally, our results suggest the importance of understanding the role of sex hormones in malaria, because they may contribute to the development of more personalized therapeutic approaches that consider patient sex, and because the oxidative stress, metabolism, hormones, and immune system in malaria differ between males and females.

## Figures and Tables

**Figure 1 ijms-26-03898-f001:**
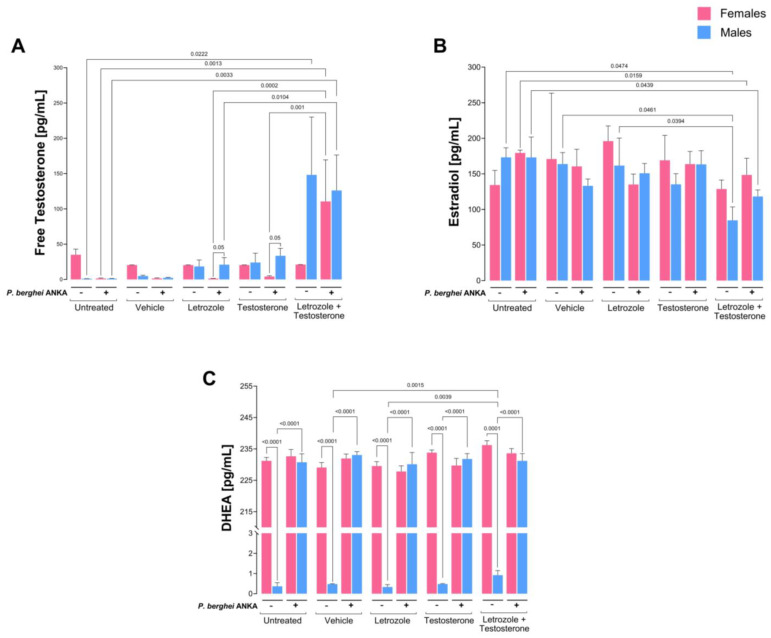
The combination of letrozole and testosterone increased free testosterone concentrations in both sexes but decreased estradiol concentrations only in male mice. Two batches of female and male CBA/Ca mice were organized into 5 groups of 10 mice of each sex, as follows: the first group was untreated, the second group was administered vehicle, the third group was treated with letrozole, the fourth group was treated with testosterone, and the fifth group was administered a combination of letrozole and testosterone. Each group was divided into two subgroups: one subgroup received PBS (−), and the other subgroup was infected with 1 × 10^3^ red blood cells parasitized with *P. berghei* ANKA (+). On day 8 post infection, the plasma concentrations of free testosterone (**A**), 17β-estradiol (**B**), and dehydroepiandrosterone (DHEA) (**C**) were quantified. Each bar represents the mean ± SEM of each group of female (pink) or male (blue) mice (n = 5). Differences were calculated via one-way ANOVA and the Bonferroni post hoc test. The lines indicate significant differences between groups with the corresponding *p* value.

**Figure 2 ijms-26-03898-f002:**
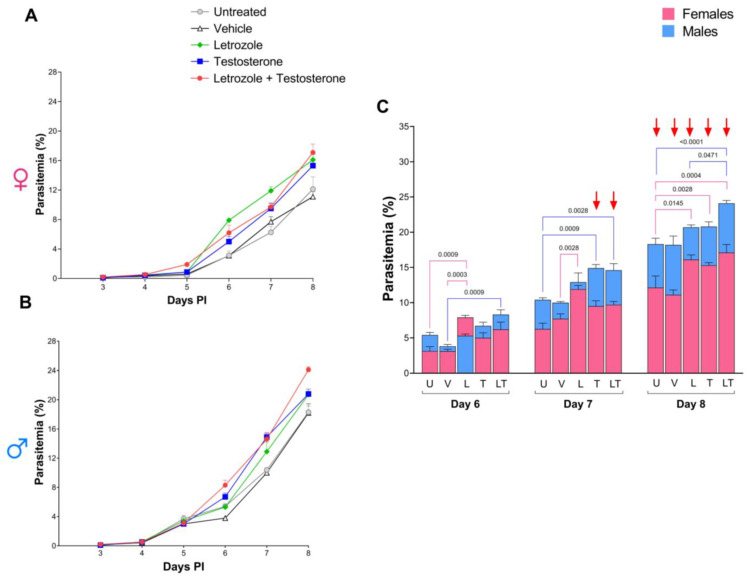
Testosterone increased parasitemia in both sexes but affected males more than females did. Two batches of female and male CBA/Ca mice were organized into 5 groups of 10 mice of each sex as follows: the first group was untreated (U), the second group was administered vehicle (V), the third group was administered letrozole (L), the fourth was administered testosterone (T), and the fifth was administered a combination of letrozole and testosterone (LT). All groups were infected with 1 × 10^3^ erythrocytes parasitized with *P. berghei* ANKA. From day 3 to day 8 post infection, parasitemia was assessed and represented as the geometric mean of the percentage of parasitized erythrocytes in the female groups (**A**) or in the male groups (**B**). Finally, (**C**) shows the parasitemia of males (blue bars) and females (pink bars) with the same treatment on days 6, 7, and 8 post infection. Each bar represents the geometric mean of the percentage of parasitemia for each group ± SEM (n = 5) Red arrows indicate difference between females and males. Differences were calculated via two-way ANOVA and the Bonferroni post hoc test. The lines indicate significant differences between groups with the corresponding *p* value.

**Figure 3 ijms-26-03898-f003:**
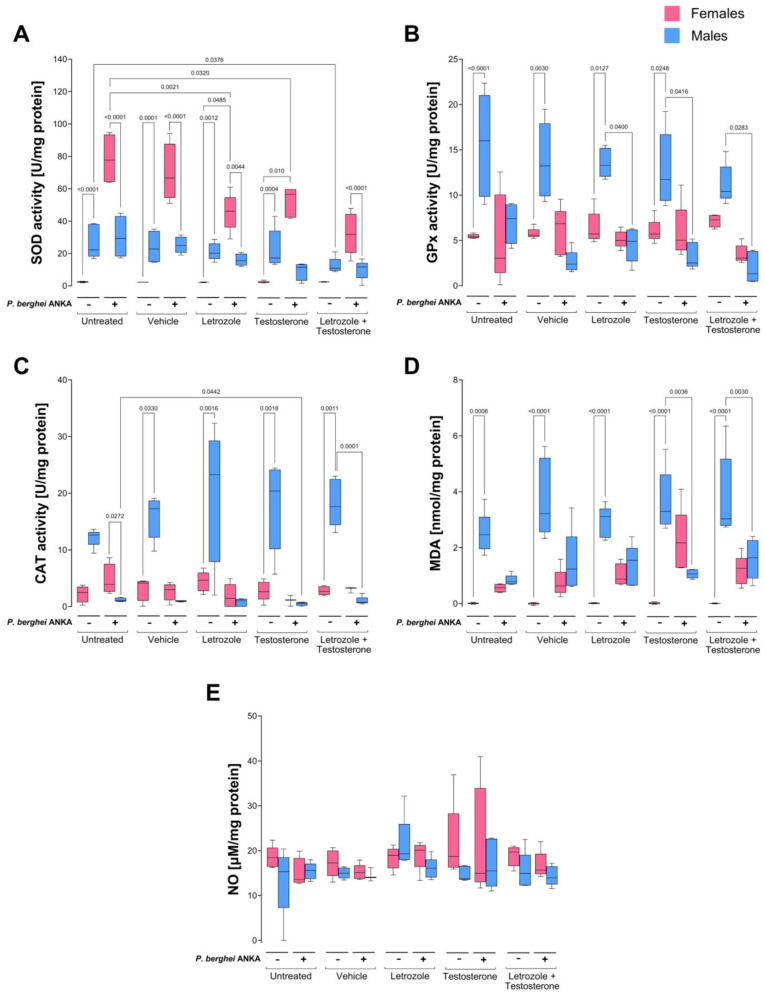
Increasing free testosterone concentrations decreased blood catalase activity in females infected with *P. berghei* ANKA. Two batches of male and female CBA/Ca mice were organized into 5 groups of 10 mice each as follows: the first group was untreated, the second group was administered vehicle, the third group was treated with letrozole; the fourth group was treated with testosterone, and the fifth group was administered a combination of letrozole and testosterone. Each group was divided into two subgroups: one subgroup received PBS, and the other subgroup was infected with 1 × 10^3^ erythrocytes parasitized with *P. berghei* ANKA. On day 8 post infection, the blood was drawn, and the enzymatic activities of superoxide dismutase (**A**), glutathione peroxidase (**B**); catalase (CAT) (**C**), the plasma malondialdehyde concentration (**D**), and the plasma NO_2_^−^ and NO_3_^−^ concentrations, which are indirect measures of nitric oxide (NO) concentration (**E**) were quantified. Each bar represents the median ± range of each group of female (pink) and male (blue) mice (n = 5). The data were analyzed via one-way ANOVA and the Bonferroni post hoc test. The lines indicate significant differences between groups with the corresponding *p* value.

**Figure 4 ijms-26-03898-f004:**
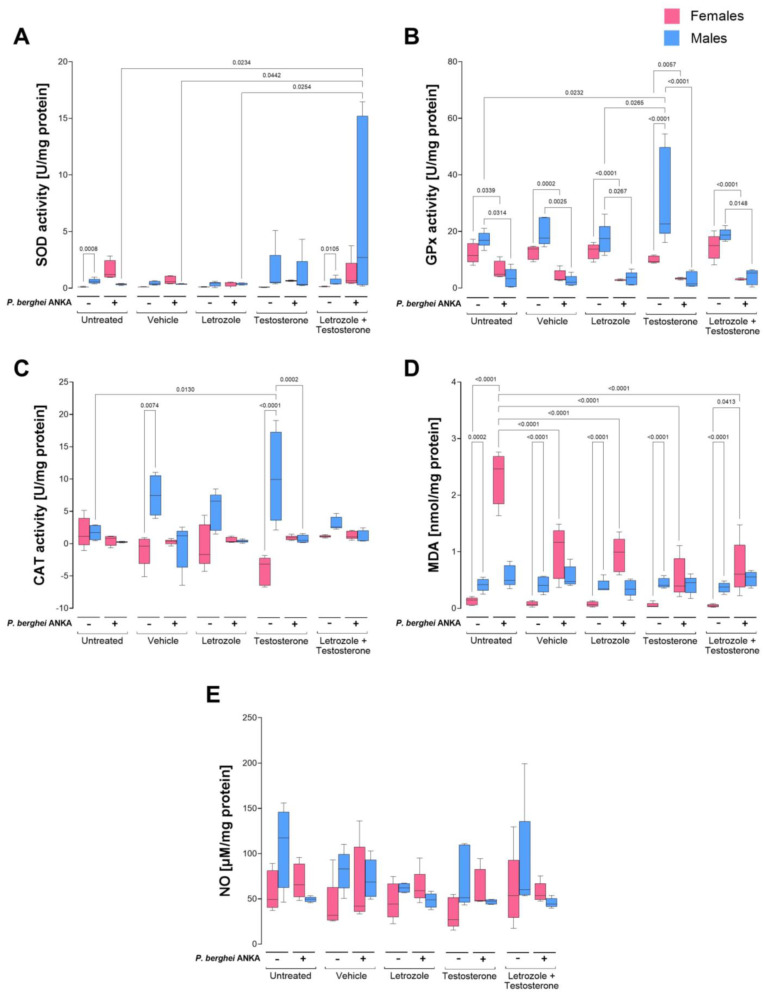
Increasing free testosterone concentrations increased superoxide dismutase activity in the spleens of male mice infected with *P. berghei* ANKA. Two batches of female and male CBA/Ca mice were organized into 5 groups of 10 mice each as follows: the first group was untreated, the second group was administered vehicle, the third group was treated with letrozole, the fourth group was treated with testosterone, and the fifth group was administered a combination of letrozole and testosterone. Each group was divided into two subgroups: one subgroup received PBS (−), and the other subgroup was infected with 1 × 10^3^ erythrocytes parasitized with *P. berghei* ANKA (+). On day 8 post infection, the spleens were removed, and the enzymatic activities of superoxide dismutase (**A**), glutathione peroxidase (**B**), and catalase (CAT) (**C**), the plasma malondialdehyde concentration (**D**), and the plasma NO_2_^−^ and NO_3_^−^ concentrations, which are indirect measures of NO concentration (**E**), were quantified. Each bar represents the median ± range for each group of female (pink) and male (blue) mice. The data were analyzed via one-way ANOVA and the Bonferroni post hoc test (n = 5). The lines indicate significant differences between the groups with the corresponding *p* value.

**Figure 5 ijms-26-03898-f005:**
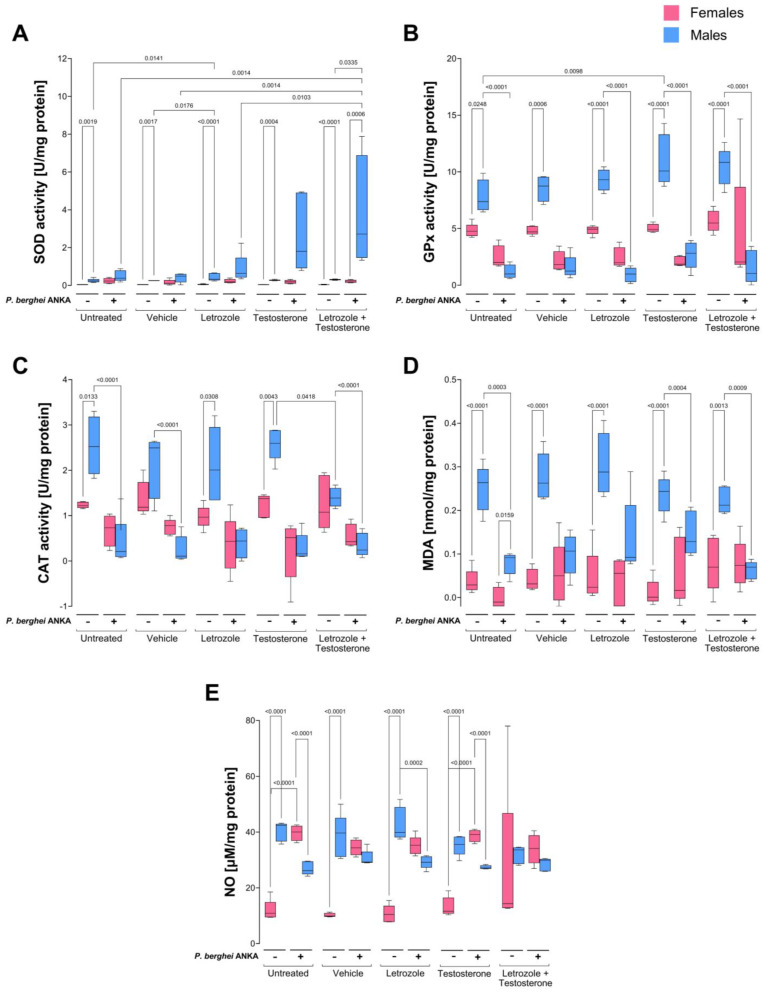
Increased free testosterone concentrations increased brain superoxide dismutase activity in *P. berghei* ANKA-infected males. Two groups of female and male CBA/Ca mice were divided into 5 groups of 10 mice each as follows: the first group was untreated, the second group was administered vehicle, the third group was treated with letrozole, the fourth group was treated with testosterone, and the fifth group was administered a combination of letrozole and testosterone. Each group was divided into two subgroups, one subgroup received PBS, and the other subgroup was infected with 1 × 10^3^ erythrocytes parasitized with *P. berghei* ANKA. On day 8 post infection, the brains were extracted and the enzymatic activities of superoxide dismutase (**A**), glutathione peroxidase (**B**), catalase (CAT) (**C**), the plasma malondialdehyde concentration (**D**), and the plasma NO_2_^−^ and NO_3_^−^ concentrations, which are indirect measures of the NO concentration (**E**), were quantified. Each bar represents the median ± range for each group of female (pink) and male (blue) mice. The data were analyzed via one-way ANOVA and the Bonferroni post hoc test (n = 5). The lines indicate significant differences between the groups with the corresponding *p* value.

**Figure 6 ijms-26-03898-f006:**
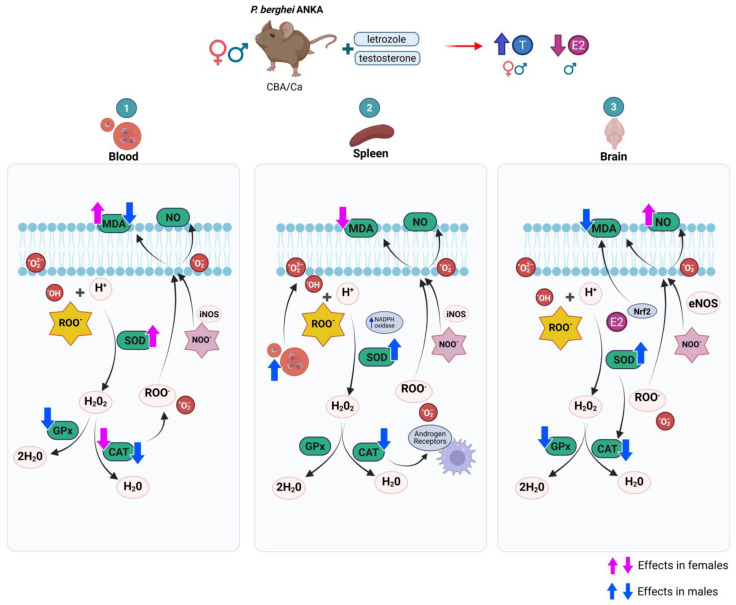
Oxidative stress caused by infection with *P. berghei* ANKA depends on testosterone concentrations, which differ between sexes and tissues. The administration of letrozole and testosterone to male and female CBA/Ca mice infected with *P. berghei* ANKA increased free testosterone concentrations in both sexes and decreased estradiol concentrations only in male mice. Increasing testosterone concentrations modified oxidative stress in the blood, spleen, and brain, which differed between the sexes. In the blood, males presented decreased SOD and GPx activities and, consequently, decreased MDA concentrations. Testosterone decreased GPx and CAT activity. In females, however, the reduction in CAT activity explains the increase in MDA. In the spleen, increasing testosterone concentrations increased SOD activity only in males, which likely neutralized superoxide anions, leaving GPx and CAT activity unchanged. The brain was the tissue that showed the greatest protection with increased testosterone concentration, particularly in males, where SOD activity increased, and MDA concentrations decreased. Moreover, nitrite and nitrate (NO) concentrations increased in females, which could protect against brain inflammation.

**Table 1 ijms-26-03898-t001:** Summary. Dimorphic effect of antioxidant enzyme activity and markers of oxidative stress in the blood, spleen, and brain in mice infected with *P. berghei* ANKA.

	Blood		Spleen		Brain	
*P. berghei* ANKA	−	+	−	+	−	+
SOD activity	♀ < ♂	↑♀ > ♂	♀ > ♂	♀ < ↑♂	♀ < ♂	♀ < ↑♂
GPx activity	♀ < ♂	♀ = ↓♂	♀ = ♂	↓♀ = ↓♂	♀ < ♂	↓♀ = ↓♂
CAT activity	♀ < ♂	♀ > ↓♂	♀ < ♂	♀ = ↓♂	♀ < ♂	♀ = ↓♂
MDA levels	♀ < ♂	↑♀ < ↓♂	♀ < ♂	↑♀ = ♂	♀ < ♂	♀ = ↓♂
NO levels	♀ = ♂	♀ = ♂	♀ = ♂	♀ = ♂	♀ < ♂	↑♀ = ♂

Uninfected (−) or infected with *P. berghei* ANKA (+) female (♀) or male (♂) mice. The upward arrows (↑) indicate an increase, and the downward arrows (↓) indicate a decrease in the specified variable in each sex of infected mice.

## Data Availability

The raw data supporting the conclusions of this paper will be made available by the authors, without reservation.

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
