# Peer review of "Testosterone Modulates Oxidative Stress in a Sexually Dimorphic Manner in CBA/Ca Mice Infected with Plasmodium berghei ANKA"

_ijms, 2025, doi:10.3390/ijms26083898_

Round 1

Reviewer 1 Report

Comments and Suggestions for Authors

Introduction

1. The introduction is quite comprehensive from a biological and conceptual standpoint; however, it could still be improved in several aspects.

2. Although the manuscript mentions that the role of testosterone is controversial, it does not clearly define the specific gap in the current knowledge. The authors are encouraged to better highlight this knowledge gap.

3. The objective of the study is presented, but no formal hypothesis is explicitly stated. It is recommended that the authors clearly formulate the hypothesis guiding their work.

4. The authors should also justify the choice of tissues analyzed (blood, spleen, and brain), especially in the context of malaria pathophysiology.

5. The clinical and translational relevance of the study should be better contextualized. For example, it would be interesting to mention that understanding the role of sex hormones in malaria may contribute to the development of more personalized therapeutic approaches that consider patient sex.

6. The authors are also encouraged to include mortality data in males and females with malaria, to reinforce the rationale for studying sex-based differences.

7. Line 60: "It has been shown that male and female mice infected with Plasmodium berghei ANKA differ in oxidative stress."Please elaborate on the nature of these differences: which tissues are affected, and through which oxidative stress pathways?

8. Line 65: Please check the abbreviation “reactive nitrogen species (NOS)”—this appears to be incorrect. Please also verify other abbreviations throughout the text.

9. Line 71: “Consequently, the heme group and Fe²⁺ are released, which oxidizes to Fe³⁺ and generates O₂⁻.” This sentence needs clarification, particularly regarding the correct mechanism of superoxide formation.

10. Line 73: Superoxide dismutase (SOD) does not act via the Fenton reaction. Please review and correct the described mechanism of SOD activity.

11. The compound letrozole should be introduced and justified in the introduction as part of the experimental design.

Results

12. Suggestion regarding Figures presenting enzymatic activity and oxidative markers:
Currently, the data for enzymatic activities—superoxide dismutase (SOD) (A, B), glutathione peroxidase (GPx) (C, D), catalase (CAT) (E, F)—as well as for oxidative stress biomarkers—plasma malondialdehyde (MDA) (G, H) and nitric oxide (NO) levels (I, J)—are presented in separate panels for non-infected and infected animals. While this approach reflects the experimental design, it makes direct visual comparison across all groups less intuitive.

13. I strongly recommend consolidating the data into single unified graphs for each variable, including all experimental groups (females and males, infected and non-infected), organized sequentially on the x-axis. For example:

  • Females (U, V, L, T, LT) → Males (U, V, L, T, LT)
    or
  • U (♀/♂), V (♀/♂), L (♀/♂), T (♀/♂), LT (♀/♂)

This format, paired with sex-specific color coding (e.g., pink for females, blue for males), would allow the reader to more easily appreciate:

  • The impact of infection within the same treatment and sex;
  • The effect of hormonal manipulation across sexes;
  • Dimorphic patterns of oxidative stress and antioxidant enzyme activity in a tissue-specific manner.

14. The authors are also requested to: Provide actual p-values instead of using general terms such as p > 0.05 or p < 0.05.

15. Review all references for correctness and formatting in accordance with ICMJE recommendations (www.icmje.org).

16. Cite original research articles instead of reviews, book chapters, or guidelines whenever possible.

Discussion

17. Although catalase activity was assessed, the discussion on this enzyme is superficial or absent in some tissues. The authors should further explore and interpret these results.

18. The discussion refers to “sexual dimorphism” in general terms, but it could be more specific in addressing the interaction between testosterone and each antioxidant enzyme in each tissue.

19. Consider including a summary table that highlights the sex-based enzymatic differences per tissue (e.g., “SOD activity increased in the brain of females but not in the spleen…”), which would help readers better integrate the findings.

Materials and Methods

20. Please justify the choice of the CBA/Ca mouse strain for this model.

21. The following citation was not found in the reference list:
“WP Benten, Bettenhaeuser, Wunderlich, Van Vliet, & Mossmann, 1991.” Please check and complete this reference appropriately.

22. Regarding nitric oxide measurement: the Griess method does not directly quantify nitric oxide (NO). Rather, it quantifies nitrite (NO₂⁻), a stable metabolite of NO, as an indirect indicator of NO production. The authors should revise all mentions of this assay accordingly and adjust any figures or text where NO is referred to as being directly measured.

Author Response

Response to Reviewer 1 Comments

Thank you very much for taking the time to review this manuscript. Please find the detailed responses below and the corresponding corrections in the resubmitted files

Point-by-point response to Comments and Suggestions for Authors

Introduction

Comment 1: The introduction is quite comprehensive from a biological and conceptual standpoint; however, it could still be improved in several aspects.

Response 1: Thank you for pointing this out, we agree with your comment. Therefore, we have addressed all of them and believe they have helped us improve and clarify the introduction. Text changes are highlighted in red.

Comment 2. Although the manuscript mentions that the role of testosterone is controversial, it does not clearly define the specific gap in the current knowledge. The authors are encouraged to highlight this knowledge gap.

Response 2: Thank you for your comment. In the revised version lines 27- 29 we have addressed this comment.

Comment 3. The objective of the study is presented, but no formal hypothesis is explicitly stated. It is recommended that the authors clearly formulate the hypothesis guiding their work.

Response 3. Thank you for your comment in the revised version we have addressed this comment in  lines 89- 92.

Comment 4. The authors should also justify the choice of tissues analyzed (blood, spleen, and brain), especially in the context of malaria pathophysiology.

Response 4. This explanation is included in the introduction to the revised version of the manuscript, lines 96- 102.

Comment 5. The clinical and translational relevance of the study should be better contextualized. For example, it would be interesting to mention that understanding the role of sex hormones in malaria may contribute to the development of more personalized therapeutic approaches that consider patient sex.

Response 5. Thank you very much for your comment; we completely agree to incorporate this suggestion in the abstract lines 37- 40, in the introduction lines 113- 114, and in the conclusions section lines 594-596.

Comment 6. The authors are also encouraged to include mortality data in males and females with malaria, to reinforce the rationale for studying sex-based differences.

Response 6. Thank you for your comment, mortality varies depending not only on sex but also on age, and if women are pregnant mortality is inverted, this is not our case. In lines 46 and 47 we assessed the mortality ratio between men and non-pregnant women. Unfortunately, differences between sexes usually are not reported; however, in the revised version, we included the rate of mortality (3:1) described in the literature.

Comment 7. Line 60: "It has been shown that male and female mice infected with Plasmodium berghei ANKA differ in oxidative stress."Please elaborate on the nature of these differences: which tissues are affected, and through which oxidative stress pathways.

Response 7. In the revised version we have addressed this comment in lines 65- 67.

Comment 8. Line 65: Please check the abbreviation “reactive nitrogen species (NOS)”—this appears to be incorrect. Please also verify other abbreviations throughout the text.

Response 8. Thank you for your comment. In the revised version, we have corrected this mistake in lines 71- 74 and 76.

Comment 9. Line 71: “Consequently, the heme group and Fe²⁺ are released, which oxidizes to Fe³⁺ and generates O₂⁻.” This sentence needs clarification, particularly regarding the correct mechanism of superoxide formation.

Response 9. Thank you very much for your comment. In the revised manuscript, we have corrected the mechanism for superoxide formation lines 80- 81.

Comment 10. Line 73: Superoxide dismutase (SOD) does not act via the Fenton reaction. Please review and correct the described mechanism of SOD activity.

Response 10. Thank you for your comment. In the revised we have described the SOD activity lines 82- 83.

Comment 11. The compound letrozole should be introduced and justified in the introduction as part of the experimental design.

Response 11. Thank you very much for your comment. We have introduced the compound letrozole in the experimental design in the introduction section on lines 94- 95.

Results

Comment 12. Suggestion regarding Figures presenting enzymatic activity and oxidative markers:

Currently, the data for enzymatic activities—superoxide dismutase (SOD) (A, B), glutathione peroxidase (GPx) (C, D), catalase (CAT) (E, F)—as well as for oxidative stress biomarkers—plasma malondialdehyde (MDA) (G, H) and nitric oxide (NO) levels (I, J)—are presented in separate panels for non-infected and infected animals. While this approach reflects the experimental design, it makes direct visual comparison across all groups less intuitive.

Response 12. Thank you very much for your comment. In the revised version, we have modified all graphics as you suggested.

Comment 13. I strongly recommend consolidating the data into single unified graphs for each variable, including all experimental groups (females and males, infected and non-infected), organized sequentially on the x-axis. For example:

•          Females (U, V, L, T, LT) Males (U, V, L, T, LT)

or

•          U (/), V (/), L (/), T (/), LT (/)

This format, paired with sex-specific color coding (e.g., pink for females, blue for males), would allow the reader to more easily appreciate:

•          The impact of infection within the same treatment and sex;

•          The effect of hormonal manipulation across sexes;

•          Dimorphic patterns of oxidative stress and antioxidant enzyme activity in a tissue-specific manner.

Response 13. Thank you very much for your recommendation. We have modified the x axis in all the graphics accordingly to your suggestion.

Comment 14. The authors are also requested to provide actual p-values instead of using general terms such as p > 0.05 or p < 0.05.

Response 14. In the revised version.We have provided the p values in all the graphics.

Comment 15. Review all references for correctness and formatting in accordance with ICMJE recommendations (www.icmje.org).

Response 15. A In the revised version we have corrected the formatting of the references

Comment 16. Cite original research articles instead of reviews, book chapters, or guidelines whenever possible.

Response 16. In the revised version, we have cited original research references whenever possible.

Discussion

Comment 17. Although catalase activity was assessed, the discussion on this enzyme is superficial or absent in some tissues. The authors should further explore and interpret these results.

Response 17. In the revised version. We have addressed this comment in lines 385-391 and 412-417.

Comment 18. The discussion refers to “sexual dimorphism” in general terms, but it could be more specific in addressing the interaction between testosterone and each antioxidant enzyme in each tissue.

Response 18. Thank you very much for your comment. We have attempted to direct the discussion toward the interaction of testosterone with each antioxidant enzyme in each tissue. However, the specific information available in the literature is scarce; nonetheless, we conducted a thorough review of other models where testosterone is measured and where the same variables are quantified; however, there is no information available for both sexes. In addition, we analyzed whether our variables relate to each other with respect to the concentration of testosterone and the information in the literature available.

Comment 19. Consider including a summary table that highlights the sex-based enzymatic differences per tissue (e.g., “SOD activity increased in the brain of females but not in the spleen…”), which would help readers better integrate the findings.

Response 19. Thank you for your comment. In the revised version, we have included Table 1 which summarizes the highlights of sex-based differences per tissue.

Material and Methods

Comment 20. Please justify the choice of the CBA/Ca mouse strain for this model.

Response 20. Thank you for your commen. This justification is included in the manuscript lines 455- 457.

Comment 21. The following citation was not found in the reference list:

“WP Benten, Bettenhaeuser, Wunderlich, Van Vliet, & Mossmann, 1991.” Please check and complete this reference appropriately.

Response 21. Thank you very much for your comment. The reference is included in the revised version of the manuscript line 814.

Comment 22. Regarding nitric oxide measurement: the Griess method does not directly quantify nitric oxide (NO). Rather, it quantifies nitrite (NO₂⁻), a stable metabolite of NO, as an indirect indicator of NO production. The authors should revise all mentions of this assay accordingly and adjust any figures or text where NO is referred to as being directly measured.

Response 22. Thank you very much for your commen. On the revised version of the manuscript we modified and adjusted the text as recommended.

Reviewer 2 Report

Comments and Suggestions for Authors

The paper is devoted to an apparently interesting topic. However, its conclusions are empiric. The mechanism(s) of regulation of SOD, CAT, GPX by 17beta-estradiol and testosterone should be more thoroughly discussed (enhanced/suppressed expression, etc.). The unclear point is the determination of NO, its concentrations seem to be significantly overestimated.  The Griess method is used for the determination of nitrate/nitrite, but not for RSNOs, which may be of equal importance. 

Minor points: #326 - NADPH reductase? Presumably, this is glutathione reductase. #389,402 - testosterone decreases eNOS activity - please explain the mechanism. 

Author Response

Response to Reviewer 2 Comments

Thank you very much for taking the time to review this manuscript. Please find the detailed responses below and the corresponding corrections in the resubmitted files.

Point-by point response to Comments and Suggestions for Authors

Comment 1: The paper is devoted to an apparently interesting topic. However, its conclusions are empiric. The mechanism(s) of regulation of SOD, CAT, GPX by 17beta-estradiol and testosterone should be more thoroughly discussed (enhanced/suppressed expression, etc.). The unclear point is the determination of NO, its concentrations seem to be significantly overestimated.  The Griess method is used for the determination of nitrate/nitrite, but not for RSNOs, which may be of equal importance.

Response 1. We appreciate your comments because they helped us to improve our manuscript. We agree with your comment about the empirical conclusions. Unfortunately, we did not measure the expression of SOD, CAT or GPx in our mice and in the literature there are conflicting results regarding whether testosterone increases or decreases the expression of the antioxidant enzymes in malaria. This is a complex phenomenon in which the expression of these enzymes varies depending on the tissue, sex, and sex hormone concentrations. It also depends on the number of receptors; to complicate matters, there are mechanisms independent of the interaction of sex hormones with their receptors; for example, 17β-estradiol has a chemical structure that confers antioxidant activity independent of its binding to the receptor. In line 409 (revised version), we highlighted that testosterone promotes the activation of the transcription factor Nrf2 in the brain; it is also likely that CAT expression in the brain can be directly influenced by regulation T cell and macrophage signaling pathways in the spleen [1]. Furthermore, testosterone probably increases mitochondrial activity [2, 3]. lines 385-387 in the revision version.

About nitric oxide, you are right, thank you very much for your comment; in the revised version, we clarified that we do not measure nitric oxide, but rather nitrites and nitrates. We considered that nitric oxide is rapidly converted to nitrite a stable metabolite of NO; therefore, the Griess reaction is an indirect measure of NO [4]. Please see lines 104, 185, 205, 224, 247, 263, 281 and 556.

  1. Wunderlich, F.; Benten, W. P. M.; Lieberherr, M.; Guo, Z.; Stamm, O.; Wrehlke, C.; Sekeris, C. E.; Mossmann, H., Testosterone signaling in T cells and macrophages. Steroids 2002, 67, (6), 535-538.
  2. Ahmad, I.; Newell-Fugate, A. E., Role of androgens and androgen receptor in control of mitochondrial function. American Journal of Physiology-Cell Physiology 2022.
  3. Dart, D. A.; Waxman, J.; Aboagye, E. O.; Bevan, C. L., Visualising androgen receptor activity in male and female mice. PloS one 2013, 8, (8), e71694.
  4. Marin, D. P.; Bolin, A. P.; de Cassia Macedo dos Santos, R.; Curi, R.; Otton, R., Testosterone suppresses oxidative stress in human neutrophils. Cell biochemistry and function 2010, 28, (5), 394-402.

Round 2

Reviewer 1 Report

Comments and Suggestions for Authors

I no longer have any concerns

Author Response

Thank you very much for taking the time to review our manuscript.

Comment 1. I no longer have any concerns

Response Comment 1. Thank you very much for your excellent review. All your comments improved our manuscript.

Reviewer 2 Report

Comments and Suggestions for Authors

There is still a problem with nitrate/nitrite concentrations (Figs. 3E, 5E) which seem to be significantly overestimated. The authors should either recalculate these data or substantiate this comparing with the data of others, e.g.  E. Nagababu, J.M. Rifkind (2007) Free Rad. Biol. Med,42; D. Tsikas (2007) J. Chromat. B. 851). 

Author Response

Response to Reviewer 2 Comments

Thank you very much for taking the time to review our manuscript. Please find the detailed responses below and the corresponding corrections in the resubmitted files

Point-by-point response to Comments and Suggestions for Authors

Comment 1. There is still a problem with nitrate/nitrite concentrations (Figs. 3E, 5E) which seem to be significantly overestimated. The authors should either recalculate these data or substantiate this comparing with the data of others, e.g. E. Nagababu, J.M. Rifkind (2007) Free Rad. Biol. Med,42; D. Tsikas (2007) J. Chromat. B. 851). 

Response to Comment 1. Thank you very much for your comment. You are correct; we reviewed the calculations and found an error in a character when transcribing the values in the graphs (we wrote mM instead of μM). In the revised version, we have corrected this mistake in Figures 3-5.

Regarding the review article by Tssikas, we believe it is an excellent review of the Griess reaction. However, the concentrations of healthy individuals correspond to humans or rats. Conversely, we measured the NO levels in mice infected with Plasmodium berghei ANKA, which we believe is one of the reasons why the concentrations differ from our results. Concerning the article by Nagababu and Rifkind, we consider chemiluminescence to be an excellent reaction for measuring NO concentration. Unfortunately, in our experience, when we attempted to quantify NO via chemiluminescence, the technique proved to be not reproducible, as our equipment does not have a robot to perform the measurements accurately at the precise time.

It is important to note that, in infectious diseases such as malaria, the enzyme that contributes most to the total concentration of NO is iNOS, whose contribution is on the order of μM (1x10-6M). This enzyme is active in macrophages and T cells [1, 2] during the immune response to malaria, and it is the one we are interested in because it is the main source of NO in our model of malaria in mice [2]. In contrast, endothelial cells generate NO concentrations from 0.0005 to 0.1 μM [3] vía eNOS, so we consider their contribution to the total NO levels to be minimal, as we reported [2]. In addition, levels of NO vary depending on the day of infection, the strain of the parasite, the host, and the tissue [4].

  1. Ahvazi, B. C.; Jacobs, P.; Stevenson, M. M., Role of macrophage–derived nitric oxide in suppression of lymphocyte proliferation during blood–stage malaria. Journal of leukocyte biology. 1995, 58, (1), 23-31.
  2. Legorreta-Herrera, M.; Rivas-Contreras, S.; Ventura-Gallegos, J.; Zentella-Dehesa, A., Nitric oxide is involved in the upregulation of IFN-γ and IL-10 mRNA expression by CD8+ T cells during the blood stages of P. chabaudi AS infection in CBA/Ca mice. International Journal of Biological Sciences. 2011, 7, (9), 1401.
  3. Chen, K.; Popel, A. S., Theoretical analysis of biochemical pathways of nitric oxide release from vascular endothelial cells. Free Radical Biology and Medicine. 2006, 41, (4), 668-680.
  4. Nahrevanian, H.; Dascombe, M. J., Nitric oxide and reactive nitrogen intermediates during lethal and nonlethal strains of murine malaria. Parasite Immunology. 2001, 23, (9), 491-501.

Round 3

Reviewer 2 Report

Comments and Suggestions for Authors

Although with significant efforts, the authors converted mM into mKM. Congratulations!